# Resensitization of Fosfomycin-Resistant *Escherichia coli* Using the CRISPR System

**DOI:** 10.3390/ijms23169175

**Published:** 2022-08-16

**Authors:** Haniel Siqueira Mortagua Walflor, Aline Rodrigues Castro Lucena, Felipe Francisco Tuon, Lia Carolina Soares Medeiros, Helisson Faoro

**Affiliations:** 1Laboratory for Applied Science and Technology in Health, Carlos Chagas Institute, Fiocruz, Curitiba 81350-010, PR, Brazil; 2Laboratory of Cell Biology, Carlos Chagas Institute, Fiocruz, Curitiba 81350-010, PR, Brazil; 3Laboratory of Emerging Infectious Diseases, Pontifícia Universidade Católica do Paraná, Curitiba 80215-901, PR, Brazil; 4Graduate Program on Bioinformatics, Federal University of Paraná, Curitiba 81520-260, PR, Brazil

**Keywords:** fosfomycin resistance, *fosA3*, CRISPR–Cas9

## Abstract

Antimicrobial resistance is a public health burden with worldwide impacts and was recently identified as one of the major causes of death in 2019. Fosfomycin is an antibiotic commonly used to treat urinary tract infections, and resistance to it in *Enterobacteriaceae* is mainly due to the metalloenzyme FosA3 encoded by the *fosA3* gene. In this work, we adapted a CRISPR-Cas9 system named pRE-FOSA3 to restore the sensitivity of a *fosA3^+^*  *Escherichia coli* strain. The *fosA3^+^*  *E. coli* strain was generated by transforming synthetic *fosA3* into a nonpathogenic *E. coli* TOP10. To mediate the *fosA3* disruption, two guide RNAs (gRNAs) were selected that used conserved regions within the *fosA3* sequence of more than 700 *fosA3*^+^  *E. coli* isolates, and the resensitization plasmid pRE-FOSA3 was assembled by cloning the gRNA into pCas9. gRNA_195 exhibited 100% efficiency in resensitizing the bacteria to fosfomycin. Additionally, the edited strain lost the ampicillin resistance encoded in the same plasmid containing the synthetic *fosA3* gene, despite not being the CRISPR-Cas9 target, indicating plasmid clearance. The in vitro analysis presented here points to a path that can be explored to assist the development of effective alternative methods of treatment against *fosA3^+^* bacteria.

## 1. Introduction

*Escherichia coli* is a Gram-negative bacillus that is a member of the gut microbiota and can be associated with human intestinal and extraintestinal diseases [1,2,3,4]. Among the main subtypes of *E. coli*, the so-called uropathogenic *E. coli* (UPECs) are the most relevant in the epidemiological context of uncomplicated urinary tract infections (UTIs), accounting for 80 to 90% of cases, mainly affecting women [5].

Fosfomycin is an antibiotic commonly used to treat UTIs and is analogous to the phosphoenolpyruvate (PEP) produced by different species of the genus *Streptomyces* [6] and by some strains of *Pseudomonas syringae* [7]. Its discovery by Hendlin et al. occurred in 1969, when the authors demonstrated its bactericidal activity against Gram-positive and Gram-negative microorganisms. Fosfomycin has a unique mechanism of action that consists of inhibiting the UDP-N-acetylglucosamine enolpyruvyl transferase (MurA) enzyme, which is responsible for the first cytosolic step in bacterial cell-wall synthesis [8].

Fosfomycin resistance occurs due to mutations in the *murA* gene, recycling of peptidoglycan instead of de novo synthesis of its precursor molecule (UDP-MurNAc), or the production of fosfomycin-inhibiting enzymes called FosA [9]. FosA (glutathione S-transferase) is a metalloenzyme that inactivates fosfomycin by catalyzing the addition of glutathione to the fosfomycin epoxide ring through a nucleophilic attack [10]. Among the various subtypes of plasmid-associated FosA enzymes, FosA3 is the most widespread acquired fosfomycin-modifying enzyme in *Enterobacteriaceae* [11,12,13,14].

Although antimicrobial resistance (AMR) is a natural phenomenon among microorganisms [15], the continuous and indiscriminate use of antimicrobials aids the selection of resistant microorganisms in a faster and more worrying way [16,17]. Increases in AMR limit the options for treating infections, resulting in yearly increases in the incidence of deaths associated with multidrug-resistant microorganisms [16,18]. This scenario will become progressively more burdensome if alternative treatments for these infections and effective antimicrobial control and stewardship programs are not established.

Recently, several studies have evaluated the ability of the CRISPR-Cas9 system to create antimicrobials that target only resistant bacteria [19,20,21,22]. Thus, when the system’s target develops resistance genes, the CRISPR-Cas9-edited bacteria undergo a resensitization process that reverts their resistant phenotype to its non-resistant form [23]. However, to date, no study has sought to evaluate these methodologies for the resensitization of fosfomycin-resistant bacteria. In this context, we aimed to adapt a system mediated by CRISPR-Cas9, called pRE-FOSA3, for the resensitization of fosfomycin-resistant *fosA3^+^ E. coli*.

## 2. Results

### 2.1. Analysis of the Occurrence of fosA Genes in Bacteria

We used genotypic data from 88,998 *fosA3^+^* bacterial isolates available in NDARO to analyze the occurrence of *fosA3* genes in *E. coli* isolates. Among them, only 4926 entries corresponded to the *E. coli* species (5.53%). FosA3 was the main fosA allele found in *E. coli*, with a relative frequency of 84.38% in Brazil and 45.5% worldwide (Figure 1).

Analysis of multiple sequence alignment revealed that the FosA3 enzyme is highly conserved among several species of clinically relevant *Enterobacteriaceae* (Appendix A). Based on this, we chose to use *fosA3* as the main target for resensitization since the other FosA proteins were not common in *E. coli* or did not have conserved segments that allowed the design of gRNAs (Appendix A). Furthermore, the data obtained between 2015 and 2020 showed a continuous increase in *fosA3^+^* isolates in terms of both total numbers and *E. coli* isolates only (Appendix A).

### 2.2. Generation of a Fosfomycin-Resistant E. coli Strain

To create a *fosA3^+^* fosfomycin-resistant *E. coli* strain, the *fosA3* allele was synthesized and cloned into the pBlueScript II SK (+) vector, generating pFOSA3 (Appendix A). The pFOSA3 vector has two resistance determinants: the *ampR* gene (ampicillin resistance), which is used as a selection marker, and the *fosA3* gene, the target of this study. The pFOSA3 vector was transformed into *E. coli* TOP10 and subjected to tests to determine the resistance phenotype. The disk diffusion tests showed the absence of inhibition halos for ampicillin and fosfomycin, confirming the fosfomycin-resistant phenotype upon acquisition of the pFOSA3 plasmid (Figure 2).

### 2.3. pFOSA3^+^ E. coli Resistance Reversal

The method used in this work allows the expression of gRNAs and the Cas9 enzyme in a single plasmid with a selective marker for chloramphenicol. Thus, gRNAs_195 and 198, which targeted the synthetic *fosA3* allele, were individually cloned into pCAS9, which contained the Cas9 enzyme sequence derived from *Streptococcus pyogenes*, resulting in pRE-FOSA3^195^ and pRE-FOSA3^198^. To test the efficiency of gRNAs in reverting the fosfomycin-resistant phenotype of the pFOSA3^+^  *E. coli* strain to its non-resistant form, we transformed the bacteria with pRE-FOSA3^195^ and pRE-FOSA3^198^ and evaluated the resistance phenotype with disk diffusion assays. The pFOSA3^+^ strain transformed with pRE-FOSA3^195^ showed inhibition halos that confirmed resistance to chloramphenicol (0 mm) and sensitivity to ampicillin (24 ± 0.5 mm) and fosfomycin (44 ± 0.5 mm), with a significant difference (*p* < 0.0001) when compared to the resistant control (pFOSA3^+^) and CRISPR control (pFOSA3^+^/pCAS9^+^) groups (Figure 3, Table 1).

In accordance with CLSI guidelines, the bacteria treated with pRE-FOSA3^138^ remained resistant to fosfomycin and did not generate statistically significant results compared to the sensitive control. However, after an additional 24 h of incubation, pRE-FOSA3^138^ reverted the resistant phenotype of pFOSA3^+^ bacteria to the non-resistant phenotype. Furthermore, the bacteria treated with pRE-FOSA3^195^ had a resensitization efficiency of 100 ± 0%, while the efficiency observed with the bacteria that received pRE-FOSA3^198^ was 4.74 ± 0.9%. Consequentially, we chose to further investigate the effects only associated with pRE-FOSA3^195^.

Growth curves for bacteria in the presence or absence of fosfomycin were analyzed to determine the sensitivity to fosfomycin. While the resistant control (pFOSA3^+^) and CRISPR control (pFOSA3^+^/pCAS9^+^) groups showed no significant difference in their growth conditions in the presence of fosfomycin (Figure 4a,b, respectively), the bacteria in the sensitive control group (pFOSA3^−^) and resensitized group (pFOSA3^+^/pRE-FOSA3^+^) (Figure 4c,d, respectively) showed similar sensitivities to each other, with a statistically significant difference (*p* < 0.0001) associated with the presence or absence of fosfomycin.

These results indicate that the resensitized bacteria underwent a reversal of the resistant phenotype in the presence of the resensitization vector pRE-FOSA3195. The observed reversal corroborates what was previously observed with disk diffusion assays. Furthermore, it is worth noting the similar growth observed in the CRISPR control, which received only Cas9 without gRNA, and the resistant control, which indicates that the Cas9 enzyme did not have a negative effect on bacterial growth even in the presence of fosfomycin.

## 3. Discussion

Fosfomycin is a relatively old antibiotic often used in clinical practice to treat lower UTIs in women [9,24,25]. Due to its unique chemical structure and low incidence of cross-resistance, fosfomycin has been explored for the treatment of infections caused by multidrug-resistant *Enterobacteriaceae* [9,24,25]. The main pathogen responsible for most lower UTIs, both in women and men, is *E. coli* [26,27]. The main form of fosfomycin resistance found in *E. coli* isolates and other *Enterobacteriaceae,* such as *Salmonella* spp. and *Shigella* spp., occurs through the production of fosfomycin-modifying enzymes belonging to the FosA family. In this study, it was observed that the *fosA3* allele was the main subtype found in *E. coli* isolates (84.38% in Brazil and 45.5% worldwide), similar to what has been shown in previous studies [12,28,29]. Furthermore, the analyses of the isolates deposited in the NDARO database in this study demonstrated that the incidence of fosfomycin resistance in bacteria is more than doubling each year. However, given the SARS-CoV-2 pandemic and the possible reduction in the efficiency of antimicrobial control, monitoring, and stewardship programs, the reported data may be underestimated [30,31,32].

Regarding the sequence of the FosA3 protein, our results showed that it is highly conserved between and within species, similar to what has already been described [12,33], reinforcing the suitability of its selection as a target for strategies involving the reversal of the fosfomycin-resistant phenotype mediated by gene disruption. Another important factor associated with *fosA3* is the presence of other determinants in the same mobile element where the *fosA3* allele is located. Several recent studies have reported the presence of *fosA3* collocated with *bla**_CTX-M_* alleles and surrounded by two IS26 segments in opposite directions in conjugable plasmids [34,35,36,37,38]. In addition, most reported cases of bacteria producing broad-spectrum beta-lactamases (ESBL) and FosA3 refer to environmental samples, such as food, river water, sewage, and carcasses [38,39]. Another recent study presented evidence of *bla_CTX-M_*^+^ and *fosA3*^+^ ESBL-producing *E. coli* in different samples associated with poultry [40]. These data, taken together, indicate that the presence of bacteria producing FosA3 and ESBL in the community context, serving as reservoirs, may be decisive for the future dissemination of these resistance abilities. Consequentially, strategies to combat fosfomycin resistance are extremely relevant.

The CRISPR-Cas9 system has been demonstrated to effectively promote new opportunities for the eradication of multidrug-resistant strains in vitro by disrupting resistance determinants or vital elements for the replication of the plasmids in which they are found [19,21,22]. Based on this, we investigated the potential of the CRISPR-Cas9 system for combatting AMR mediated by the *fosA3* allele present within a commercial high-copy plasmid, pBC-SK II, which was transformed into a nonpathogenic *E. coli* TOP10 strain. To achieve this, two regions fully conserved in all 716 *fosA3^+^ E. coli* isolates, which were present in a relatively early region of the gene, were selected in order to design two gRNAs (gRNA_195 and gRNA_198) for further resensitization assays.

pRE-FOSA3^195^ was able to completely reverse the resistant phenotype of pFOSA3^+^ bacteria, similar to what has been achieved with other types of resistance [19,21,22]. Furthermore, pRE-FOSA3^195^ was more efficient than pRE-FOSA3^198^. pRE-FOSA3^198^ was able to resensitize pFOSA3^+^ bacteria after only two days, approximately, of total incubation time [41]. The CHOP-CHOPv3 prediction program showed that these were the two most efficient gRNAs (Appendix A). Additionally, the distance between the two gRNAs is only three nucleotides, and the GC content is 5% higher in gRNA_198. We do not know the reason for such a significant difference in the efficiency of resensitization. However, the importance of investigating two or more gRNAs when intending to use CRISPR-Cas9 is evident, given that the efficiencies calculated by the prediction programs are only estimated and are not necessarily reproduced in in vitro or in vivo assays.

The discrepancy observed may have been associated with the significant variation in the action of the ribonucleoprotein complex (RNP) between different types of target sites or cell lines, involving several factors that can influence the binding and cleavage efficiency of the RNP complex [42,43,44,45,46,47,48,49,50]. Considering this in addition to the fact that most prediction programs, including the one used in this work, base their scores on empirical decisions using different computational methods, these programs should be considered as experimental bases for the selection of sets of gRNAs, allowing their efficiency and the variables in question to be tested in study models [47,50].

Nevertheless, in addition to the reversal of the fosfomycin-resistant phenotype in the bacteria treated with pRE-FOSA3^195^, we observed the reversal of the ampicillin resistance encoded in the *ampR* gene collocated on the same plasmid as *fosA3* (pFOSA3). This suggests that, after the disruption of *fosA3*, *E. coli* TOP10 pFOSA3^+^ possibly suffered the elimination of the pFOSA3 plasmid, thus losing the determinants of resistance to both fosfomycin and ampicillin. This phenomenon has been described previously [19,21] and can be used as a strategy to increase the efficiency of resensitization or even to reverse the phenotypes of several types of resistance with sequences located on the same plasmid. Although we applied this strategy only to *E. coli* strains, other studies have explored the same principles with other bacteria species, such as *Salmonella enterica*, *Klebsiella pneumoniae*, *Enterobacter hormaechei*, *Enterobacter xiangfangensis*, *Serratia marcescens* [51], and methicillin-resistant *Staphylococcus aureus* [21]. This indicates that using CRISPR to kill or resensitize bacteria has potential for broad application with different pathogenic microorganisms and resistance determinants. However, it is important to note that the ideal CRISPR target will depend on the characteristics inherent to the microorganism in question, and the technique’s success will also imply a good choice of target and a delivery method suitable for the desired microorganism.

Despite the evident results for the in vitro resensitization of the fosfomycin-resistant bacteria obtained in this work, no tests have been carried out on clinical isolates of fosfomycin-resistant *E. coli* to date. Thus, to bring more robustness to what has been developed, evaluation of the pRE-FOSA3 system in different sets of fosfomycin-resistant isolates remains a prospective approach that could be used to assess whether this system has the efficiency and replicability necessary for the development of a strategy to combat AMR. Furthermore, despite being an in vitro study, in vivo evaluations and exploration of pRE-FOSA3 delivery strategies would be extremely important to solidify the results obtained in this work. Finally, we hope that the target sequences determined in this study will provide a basis for developing strategies to combat *fosA3*-mediated fosfomycin resistance.

## 4. Materials and Methods

### 4.1. Analysis of the Occurrence of fosA Genes in Bacteria

Analysis of the occurrence of *fosA* genes in bacteria was performed using data from the National Database of Antibiotic-Resistant Organisms (NDARO). The keywords “fosA” and “fosfomycin resistance” were used to acquire data with the Pathogen Browser search tool. A .csv file containing data on 89,998 isolates of various bacterial species was obtained and analyzed in Python to generate representative graphs of the occurrence of *fosA* genes in *E. coli* isolates.

### 4.2. Analysis of the fosA3 DNA and Protein Sequences

A .csv file with data regarding the nucleotide sequences of the *fosA3* allele was obtained with the Pathogen Detection Microbial Browser for Identification of Genetic and Genomic Elements (MicroBIGG-E) from the National Center for Biotechnology Information (NCBI, NIH). The contig identification sequences of each *fosA3* gene were used to recover the *fosA3* gene sequences from 716 *E. coli* isolates that showed identities greater than 90% with the reference sequence (WP_014839980.1). The sequences were aligned with MUSCLE in Jalview software [52,53]. The reference FosA sequences are listed in Appendix A, and the alignment can be seen in Appendix A.

### 4.3. Design of gRNAs Targeting fosA3

The alignment of the 716 *fosA3* sequences was used to choose fully conserved regions for the design of gRNAs. The prediction of gRNA was performed in the online web server CHOP-CHOPv3 (http://chopchop.cbu.uib.no/) accessed on 27 May 2021 [41] using the consensus sequence of the *fosA3* generated in the alignment. The genome of the uropathogenic strain *E. coli* CF7073 (ASM744v1) was used as a reference for off-target estimation. A .csv file with the results was obtained and filtered to show only gRNA sequences that showed no off-target effects with the reference genome and targeted regions without mutations over a 20 nucleotide span. In all, 61 gRNAs were obtained. Two gRNAs were chosen for the analyses (gRNA_195 and gRNA_198) based on the efficiency score and annealing position provided by CHOP-CHOPv3 (Appendix A).

### 4.4. pRE-FOSA3 Assembly

The pRE-FOSA3 plasmid was constructed based on the pCas9 vector purchased from Addgene (#42876). gRNAs 195 and 198, targeting *fosA3*, were commercially synthesized (Exxtend, SP, BR) and cloned into the pCas9 vector, adapting the BsaI enzyme protocol [54]. At the end of the cloning, two vectors expressing Cas9 and gRNA_195 (pRE-FOSA3^195^) or gRNA_198 (pRE-FOSA3^198^) were obtained.

### 4.5. Generation of an E. coli Fosfomycin-Resistant Strain

To construct a fosfomycin-resistant strain of *E. coli*, the *fosA3* gene was synthesized (FastBio, São Paulo, Brazil) in the pBlueScript II SK (+) vector using a cloning strategy with BamHI and HindIII. The product of this cloning was a vector that expressed the *fosA3* gene constitutively, with a selection marker for ampicillin (*ampR*) named pFOSA3 (Appendix A). pFOSA3 was transformed by heat shock into chemically competent *E. coli* TOP10. Then, the transformed cells were plated on LB agar containing 100 μg/mL ampicillin. The plate was incubated at 37 °C overnight. To confirm the fosfomycin-resistant phenotype, disk diffusion assays were performed.

### 4.6. Resensitization of E. coli pFOSA3^+^ with pRE-FOSA3

To deliver pRE-FOSA3 vectors to *E. coli* pFOSA3^+^ by heat shock, approximately 100 ng of each vector (pRE-FOSA3^gRNA_195^ or pRE-FOSA3^gRNA_198^) was added separately to tubes containing chemically competent pFOSA3^+^ cells. Cells were recovered in lysogeny broth (LB) without antibiotics at a dilution of 1:10 over 1 h at 37 °C and under agitation at 200 rpm. Afterward, 100 μL of the contents of each tube was plated on two LB plates under different conditions: (1) nonselective plate—LB + chloramphenicol (34 μg/mL); (2) selective plate—chloramphenicol (34 μg/mL) + glucose-6-phosphate (25 μg/mL) + fosfomycin (40 μg/mL). As a sensitive control, an isolated colony of *E. coli* TOP10 was submitted to the same procedures described above, except for pRE-FOSA3 delivery. Resensitization efficiency was calculated as follows: (1—number of colonies on the selective plate/number of colonies on the nonselective plate × 100%).

### 4.7. Disk Diffusion

The fosfomycin-resistant phenotype of *fosA3*^+^  *E. coli* was determined using fosfomycin disks (200 μg) supplemented with glucose-6-phosphate (50 μg). Ampicillin disks (10 μg) were used to determine the presence of pFOSA3. After resensitization, the resistance phenotype was assessed using disks of fosfomycin (200 μg) supplemented with glucose-6-phosphate (25 μg), ampicillin (10 μg), and chloramphenicol (30 μg) with the colonies isolated from the nonselective plates (LB + 34 μg/mL chloramphenicol). These disk diffusion assays were performed in technical duplicates, and the entire experiment was repeated three times to account for the biological replicates in accordance with CLSI guidelines.

### 4.8. Growth Curve

Three preinocula were produced using colonies isolated from the nonselective plates of each group in LB with or without antibiotics as follows: (1) *E. coli* pFOSA3^+^—ampicillin (100 μg/mL); (2) *E. coli* pFOSA3^+^/pCAS9^+^—ampicillin (100 μg/mL) and chloramphenicol (34 μg/mL); (3) *E. coli* pFOSA3^+^/pRE-FOSA3^+^—chloramphenicol (34 μg/mL); and (4) *E. coli* TOP10 (pFOSA3^–^)—no antibiotics. Each preinoculum was incubated at 37 °C with shaking at 200 rpm overnight. The following day, they were adjusted to 0.5 McFarland standard, and 75 μL of each adjusted preinoculum was added to a 96-well microplate containing LB with or without fosfomycin (40 μg/mL) + glucose-6-phosphate (25 μg/mL). Growth kinetics were recorded every hour for 24 h. Each assay was performed in technical triplicate, and the entire experiment was repeated three times for the biological replicates.

### 4.9. Statistical Analysis

Statistical analyses were assessed using GraphPad Prism software (version 9, GraphPad Software, California, United States) and Python 3.8.8. For both the resensitization and disk diffusion assays, the inference of the significance was made using an unpaired t-test with Welch correction.

## Figures and Tables

**Figure 1 ijms-23-09175-f001:**
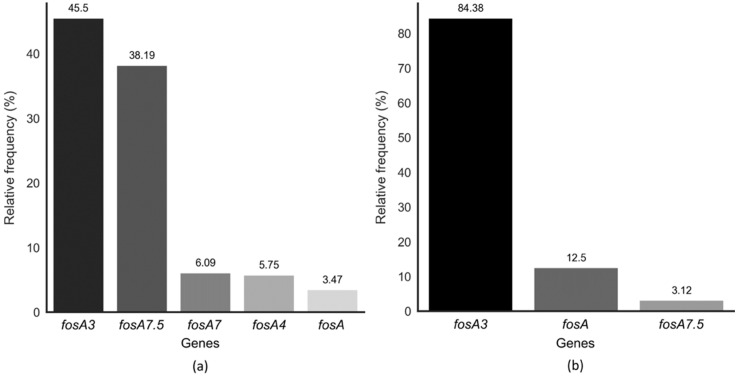
Occurrence of *fosA* alleles in *Escherichia coli* isolates. (**a**) Worldwide isolates; (**b**) isolates located in Brazil.

**Figure 2 ijms-23-09175-f002:**
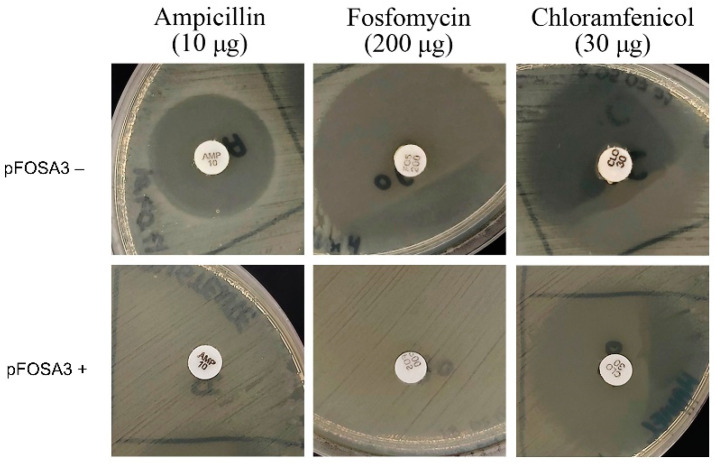
Confirmatory disk diffusion of the resistant phenotype of pFOSA3+ *E. coli* TOP10. Representative photographs of disk diffusion plates obtained in one of the assays.

**Figure 3 ijms-23-09175-f003:**
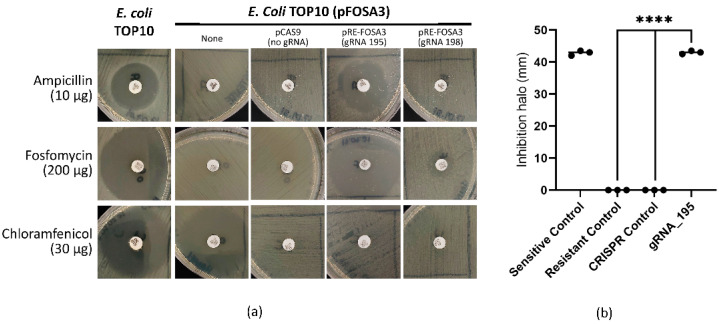
Disk diffusion test with *E. coli* grown on Mueller–Hinton agar. (**a**) Representative photographs of disk diffusion plates obtained in one of the assays showing the presence/absence of inhibition halos. (**b**) A graph showing the data obtained from the inhibition halos, with fosfomycin disks (μg) supplemented with glucose-6-phosphate; data were obtained in biological and technical replicates; black dots represent individual measurements of the inhibition halos, and asterisks indicates significant difference **** *p* < 0.0001. Sensitive control: TOP10/pFOSA3^--^; CRISPR control: TOP10/pFOSA3^+^/pCAS9^+^; Resistant control: TOP10/pFOSA3^+^; gRNA_195: TOP10/pFOSA3^+^/pRE-FOSA3^gRNA_195^. The inference was made with an unpaired *t*-test with Welch correction.

**Figure 4 ijms-23-09175-f004:**
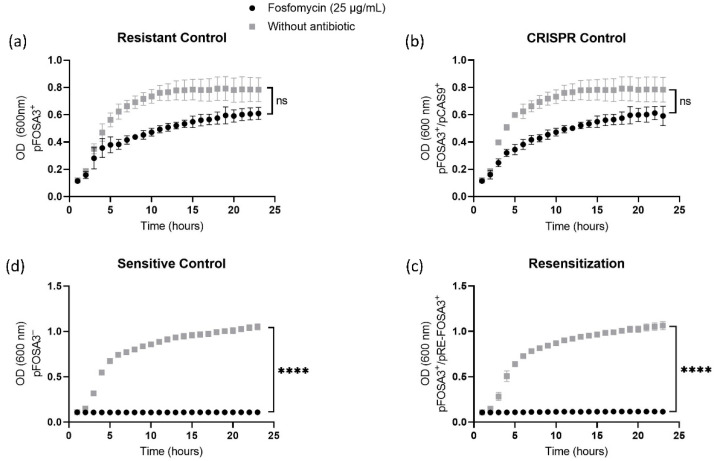
Growth curves for bacteria in medium with or without fosfomycin (supplemented with 25 μg/mL glucose-6-phosphate). (**a**) Resistant control (pFOSA3^+^); (**b**) CRISPR control (pFOSA3^+^/pCAS9); (**c**) sensitive control (pFOSA3^–^); and (**d**) resensitization group (pFOSA3^+^/pRE-FOSA3^+^). G6P: glucose-6-phosphate. Asterisk: statistically significant difference (*p* < 0.0001); ns: nonsignificant difference. Values represent the mean ± standard deviation of three independent experiments (biological and technical triplicates). The inference was made using an unpaired t-test with Welch correction. *p* < 0.0001.

**Table 1 ijms-23-09175-t001:** Inhibition-zone diameters for ampicillin, fosfomycin, and chloramphenicol.

Antibiotics	Diameter Zone (mm)				
	*E. coli* TOP10	*E. coli* TOP10 (pFOSA3^+^)			
		None	pCAS9 (Without gRNA)	pRE-FOSA3 (gRNA_195)	pRE_FOSA3 (gRNA_198) **
Ampicillin (10 μg)	23 (±0.29)	0	0	24 (±0.5)	0
Fosfomycin (200 μg) *	42 (±0.58)	0	0	44 (±0.5)	15
Chloramphenicol (30 μg)	34 (±0.86)	33 (±0.29)	0	0	0

* Disk supplemented with glucose-6-phosphate (25 μg); ** no replicates were performed. Values represent the mean ± standard deviation of biological and technical triplicates, and the inference was made with an unpaired *t*-test.

## Data Availability

Not applicable.

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
