# Peer review of "Resensitization of Fosfomycin-Resistant *Escherichia coli* Using the CRISPR System"

_ijms, 2022, doi:10.3390/ijms23169175_

Round 1

Reviewer 1 Report

The authors describe an original method using the CRISPR system to re-sensitize resistance to fosfomycin in E. coli

Minor comments:

1) What were the statistical analyzes they performed for the different experiments? please add this information.

2) Gene names should be in italics. Please fix it in the figures and in the text.

3) The alignment of protein sequences of proteins shown in table s2 should be added, indicating the differences between these sequences.

4) Could this method be applied to re-sensitize other resistances and/or in other pathogenic microorganisms? This should be more discussed.

Author Response

Minor comments:

1) What were the statistical analyzes they performed for the different experiments? Please add this information.

HF: Thanks for the note. We added the following declaration to materials and methods: “For both the resensitization and disk diffusion assays, the inference of the significance was made using the unpaired t-test with Welch correction.”

2) Gene names should be in italics. Please fix it in the figures and in the text.

HF: Text and figures have been corrected.

3) The alignment of protein sequences of proteins shown in table s2 should be added, indicating the differences between these sequences.

HF: We have added protein alignments as supplemental figure S5.

4) Could this method be applied to re-sensitize other resistances and/or in other pathogenic microorganisms? This should be more discussed.

HF: This is a very interesting point, and we appreciate the opportunity to clarify it. We added the following statement to the discussion: “Although we applied this strategy only to E. coli strains, other studies have explored the same principles with other bacterial species such as Salmonella enterica, Klebsiella pneu-moniae, Enterobacter hormaechei, Enterobacter xiangfangensis, Serratia marcescens [51], and Methicillin-Resistant Staphylococcus aureus [21]. This indicates that using CRISPR to kill or resensitize bacteria has broad application potential in different pathogenic microorganisms or resistance determinants. However, it is important to note that the ideal CRISPR target will depend on the characteristics inherent to the microorganism in question, and the technique’s success will also imply a good choice of the target and delivery method suitable for the desired microorganism.”

Reviewer 2 Report

The manuscript by Walflor et al. titled "Resensitization of fosfomycin-resistant Escherichia coli using the CRISPR system" presents a method for combating fosfomycin-resistance in Escherichia coli. As the authors themselves point out, this research is very important and a significant contribution to the development of effective alternative treatments against antibiotic-resistant bacteria. The manuscript is well structured and written, the methodology is described in detail, and the results are presented simply and clearly. The Discussion of the results is quite concise but reflects well the current state of knowledge regarding the issue addressed. Although a significant weakness of the conducted research is the work only in vitro, on prepared strains, the results obtained provide a solid basis for continuing research on clinical isolates of E. coli resistant to fosfomycin and what may be the beginning of developing strategies to combat antimicrobial resistance in bacteria. 

Author Response

The manuscript by Walflor et al. titled “Resensitization of osfomycin-resistant Escherichia coli using the CRISPR system” presents a method for combating osfomycin-resistance in Escherichia coli. As the authors themselves point out, this research is very important and a significant contribution to the development of effective alternative treatments against antibiotic-resistant bacteria. The manuscript is well structured and written, the methodology is described in detail, and the results are presented simply and clearly. The Discussion of the results is quite concise but reflects well the current state of knowledge regarding the issue addressed. Although a significant weakness of the conducted research is the work only in vitro, on prepared strains, the results obtained provide a solid basis for continuing research on clinical isolates of E. coli resistant to fosfomycin and what may be the beginning of developing strategies to combat antimicrobial resistance in bacteria. 

HF: Thanks a lot for the suggestion. I would also like to thank you for the words of encouragement.

Reviewer 3 Report

Antimicrobial resistance is a significant public health problem responsible for the leading causes of death. Fosfomycin is an antibiotic commonly used to treat urinary tract infections, and its resistance in Enterobacteriaceae is mainly due to the metalloenzyme FosA3 encoded by the fosA3 gene. In this work, the authors adapted a CRISPR ‒ Cas9 system named pRE-FOSA3  to restore sensitivity and reduce the resistance of a fosA3 + strain of Escherichia coli, obtaining promising results. The introduction seems discursive, and the materials, methods, and results are well written.

Author Response

Antimicrobial resistance is a significant public health problem responsible for the leading causes of death. Fosfomycin is an antibiotic commonly used to treat urinary tract infections, and its resistance in Enterobacteriaceae is mainly due to the metalloenzyme FosA3 encoded by the fosA3 gene. In this work, the authors adapted a CRISPR ‒ Cas9 system named pRE-FOSA3  to restore sensitivity and reduce the resistance of a fosA3 + strain of Escherichia coli, obtaining promising results. The introduction seems discursive, and the materials, methods, and results are well written.

HF: We appreciate the suggestions.